# Blocking the Bromodomains Function Contributes to Disturbances in Alga *Chara vulgaris* Spermatids Differentiation

**DOI:** 10.3390/cells9061352

**Published:** 2020-05-29

**Authors:** Agnieszka Wojtczak

**Affiliations:** Institute of Experimental Biology, Department of Cytophysiology, University of Lodz, Faculty of Biology and Environmental Protection, 141/143 Pomorska, 90-236 Lodz, Poland; agnieszka.wojtczak@biol.uni.lodz.pl; Tel.: +48-42-6354734

**Keywords:** bromodomain, *Chara vulgaris*, chromatin, chromatin remodeling complex, electron microscopy, endoplasmic reticulum stress, spermatogenesis, spermiogenesis

## Abstract

Bromodomain containing (BRD) proteins play an essential role in many cellular processes. The aim of this study was to estimate activity of bromodomains during alga *Chara vulgaris* spermatids differentiation. The effect of a bromodomain inhibitor, JQ1 (100 μM), on the distribution of individual stages of spermatids and their ultrastructure was studied. The material was Feulgen stained and analysed in an electron microscope. JQ1 caused shortening of the early stages of spermiogenesis and a reverse reaction at the later stages. Additionally, in the same antheridium, spermatids at distant developmental stages were present. On the ultrastructural level, chromatin fibril system disorders and significantly distended endoplasmic reticulum (ER) cisternae already at the early stages were observed. Many autolytic vacuoles were also visible. The ultrastructural disturbances intensified after prolonged treatment with JQ1. The obtained data show that JQ1 treatment led to changes in the spermatid number and disturbances in chromatin condensation and to cytoplasm reduction. The current studies show some similarities between *C. vulgaris* and mammals spermiogenesis. Taken together, these results suggest that JQ1 interferes with the spermatid differentiation on many interdependent levels and seems to induce ER stress, which leads to spermatid degeneration. Studies on the role of bromodomains in algae spermiogenesis have not been conducted so far.

## 1. Introduction

Spermiogenesis is one of the most complex and highly specialized morphogenetic processes. In alga *Chara vulgaris,* this process is the second, following the proliferative phase, stage of spermatogenesis. It lasts 7 days during which 10 stages (I–X) of different duration times are distinguished. During each of the stages, there are characteristic ultrastructural changes [1] that are the result, among others, of the presence of double-strand DNA breaks that allow the exchange of histones to protamines, conditioning correct chromatin remodeling [2,3].

Bromodomains, 110-amino-acid domains, are present in many chromatin-associated proteins, histone acetyltransferase and subunits of ATP-dependent chromatin remodeling complexes [4,5]. Research on bromodomains and extra-terminal (BET) family proteins has mainly included yeast and animals (*Drosophila*, human and mouse). These proteins have two bromodomains. In mammals in BET proteins, four members of bromodomain containing (BRD) proteins: BRD2, BRD3, BRD4, and BRDT (bromodomain testis-specific, called BRD6) were distinguished [6,7,8]. Analyses concerning BRD proteins in plants are in the minority in comparison to those in animals, however, they have also been revealed in plants, but they have one bromodomain [9]. Plants studies have mainly focused on *Arabidopsis*, in which 29 BRD proteins have been identified [10] and functions of only three genes, from among 12, have been characterized [11]. *Brachypodium* [12], tomato and tobacco species [13] as well as soybean [10] have also been analyzed. Although researches related to plant bromodomains are ongoing, there is still no evidence to support the Florence and Faller [14] hypothesis explaining the presence of one bromodomain in plant proteins instead of two, as is the case in animals. Based on detailed analysis, it was shown that bromodomains in plants were more similar to bromodomain 2 than bromodomain 1 present in animals [14].

BRD proteins play an important role, among others, in controlling leaf development [15], in the transcription, DNA repair and chromatin structure reorganization processes also associated with the removal of nucleosomes [4,16,17,18]. Proteins belonging to Swi2/Snf2 family, which are present in one of the chromatin remodeling complexes, also have a bromodomain motif on C-terminal region. According to the literature data, algae homologues of the SNF2 subunit have been detected so far in a few of Rhodophyta (*Cyanidioschyzon merolae*, *Porphyra* and *Chondrus*), Chlorophyta (*Chlamydomonas reinhardtii*) and recently in Charophyta (*Chara vulgaris*) [19]. There is little information about BRD proteins in algae. Presence of these kinds of proteins was shown in *Cyanidioschyzon merolae* (nine proteins) [10], *Micromonas pusilla* (strain CCMP1545, two proteins) [20], *C. vulgaris* (Brg1) [19], and in *C. reinhardtii* where the protein is characterized by the presence of as many as three bromodomains [10]; in addition, in diatom *Thalassiosira pseudonana* (strain CCMP1335) bromodomains in 27 predicted proteins were identified [21].

Inhibitors for the BET family of bromodomains have therapeutic potential and researches are conducted on many animal cancer cell models (human, mouse) [22]. Among the numerous bromodomain inhibitors, the JQ1 (thieno-triazolo-1,4-diazepine) is specific and often used in research as an anticancer drug. Studies on human and murine models showed that its biochemically active stereoisomer, (+)-JQ1, was a potent, more-specific inhibitor of BRD4 protein than of BRD2 and BRD3 [22,23]. Studies on mouse spermatogenesis revealed that bromodomain played an essential role in this process, and the loss of the first bromodomain of the *Brdt* gene caused sterility of these mammals [24,25]. In immunofluorescence studies during murine spermatogenesis, no BRD protein (BRD2, BRD4, BRDT) in condensing spermatids was demonstrated [26]. Bromodomains bind acetylated lysines, which are present in various proteins i.e., histones playing an important role in chromatin organization during spermiogenesis [27]. The results of previous immunofluorescent and ultrastructural analyses showed that blocking the removal of histones, during the exchange of these proteins into protamines, hindered the proper course of spermiogenesis [3,28]. The studies of spermatogenesis concerning *Brdt* gene in two fish species, which have different methods of nuclear protein exchange, revealed variations in this gene expression which could indicate a different role of Brdt protein [29]. The exchange of nucleoproteins in one of these species (*Dicentrarchus labrax*) follows the same scheme as in *Chara* [30]. Therefore, it is interesting how the blocking of bromodomains will affect spermiogenesis in this alga. *C. vulgaris* is a model organism which was earlier applied in the study i.e., on different processes during spermatogenesis [19]. This alga belongs to Charophyta, which are closely related to land plants [31,32].

The aim of the current work was to find out what role bromodomains play in the course of spermatid differentiation in this alga. The present research focused on whether and to what extent blocking the activity of bromodomains under the influence of their inhibitor, JQ1, affects the distribution of individual stages of spermatids and the ultrastructure of spermatids during spermiogenesis. To the best of my knowledge, this paper presents the first research on bromodomains in algae spermiogenesis.

## 2. Materials and Methods

### 2.1. Material

The research material was antheridia of *Chara vulgaris* from III–V node pleuridia (counting from the apical buds). The algae were grown in an artificial pond located in the Rogów Arboretum (Poland). Prior to the studies, the algae were grown for a few days in tanks containing water from the natural environment at room temperature under natural light.

### 2.2. Treatment with Bromodomain Inhibitor

The thallus fragments carrying antheridia were treated for 24 h and 48 h with a bromodomain inhibitor, (+)-JQ1 (hereafter referred to as JQ1) (Sigma, SML1524) at 100 μM concentration. The solution of the inhibitor was prepared with the water from the natural environment, with DMSO (Sigma) in the ratio 20 mg/mL. These duration times of treatment with the bromodomain inhibitor were used, because previous analyses [3,28] were also based on the same scheme. In the studies on animal material, using cell lines, a broad concentration range of bromodomain inhibitors, but lower than in this alga, was used. Because in the case of *C. vulgaris*, antheridial filaments were developing inside the multicellular antheridium, the concentration of the inhibitor used was many times higher in comparison with the cell line model. Initially the concentration of 50 μM was used, however, barely noticeable changes appeared only in single spermatids.

### 2.3. Feulgen Staining and Light Microscopy Studies

The inhibitor-treated material and the control, treated only with DMSO, were fixed in Carnoy solution (ethanol:glacial acetic acid, 3:1, *v/v*) for 1 h, rinsed in 96% ethyl alcohol, and kept in 70% ethyl alcohol. Next, the fixed specimens were stained with the Schiff’s reagent (Feulgen staining) according to the standard method (e.g., [33]). Next, antheridia from the treated material and the control were isolated. Squashed preparations from these antheridia were made on dry ice and then embedded in Canada balsam. The analysis of all the preparations in a light microscope was carried out in order to determine the transitory stage between proliferation phase and spermiogenesis (64/sp) and spermiogenesis stages (I–X). The percentage of spermatids at each stage of spermiogenesis, from both the control and JQ1-treated material, was calculated as a share of these spermatids from particular antheridia in the whole pool in a representative group regarded as 100%. After Feulgen staining from the material treated for 48 h with JQ1 too few antheridia were obtained, quantitative analysis of spermiogenesis stages was not possible. Because some spermatids at stage X revealed differences in the Feulgen staining intensity in comparison to the control, cytophotometric measurements were carried out. Analysis of images (in arbitrary units) was performed using a Jenamed 2 microscope (Carl Zeiss, Germany) with the computer-aided Cytophotometer v1.2 (Forel, Poland).

### 2.4. Electron Microscopy

Twenty-four hour and 48 h JQ1-treated material and the control were fixed with 3% glutaraldehyde in 0.1 M cacodylate buffer (pH 7.3) supplemented with 0.007 M CaCl2 for 3 h. The isolated antheridia were gently squashed in a drop of heated 2% agar in cacodylate buffer and postfixed in 1% OsO_4_ in the same buffer for 2 h. After dehydration in an alcohol series, the material was embedded in Epon 812 and Spurr mixture medium (Polysciences) according to the standard procedure presented earlier [3,34]. Ultrathin sections were double-stained with uranyl acetate and lead citrate according to Reynolds [35]. All ultrathin sections were examined and photographed in a JEOL JEM 1010 transmission electron microscope (TEM) at 80 kV acceleration voltage.

## 3. Results

### 3.1. A Relative Duration of C. vulgaris Stages of Spermiogenesis after Treatment with JQ1

Analyses concerned the transitory stage between proliferation phase and spermiogenesis (64/sp) and spermiogenesis stages (I–X) both in the control and JQ1, a bromodomain inhibitor, treated material. Following 48 h treatment with JQ1, thallus differed in appearance from the control one; however, after acid hydrolysis and Feulgen staining, too few antheridia were obtained from this material, therefore, quantitative analysis of spermiogenesis stages was not possible. The control frequency of spermatids at the stages I–III is greater than in V–VIII ones, and it is connected with a different duration of spermiogenesis stages [36]. Early stages last much longer in comparison with the successive ones. Comparative analysis of the percentage of spermatids in the control samples and after 24 h JQ1 treatment revealed shortening of the 64/sp stage and early stages (I–III) of spermiogenesis caused by this inhibitor, since a decrease in the number of spermatids was observed (Figure 1).

On the other hand, an increase in the number of spermatids in subsequent stages of spermiogenesis V–VIII was observed. At stage IV and IX, slight decreases in the number of spermatids in the material treated with JQ1, with a slight increase of X at the last stage, were revealed (Figure 1). Light microscope analysis of the spermiogenesis stages in the material treated with JQ1 showed differences in the Feulgen staining intensity of antheridia and some spermatids compared to the control. These differences were most visible at stage X (Figure 2C vs. Figure 2F). The cytophotometric measurements of the Feulgen staining intensity in these spermatids revealed twofold lower values (in arbitrary units).

In antheridial filaments in the control material (Figure 2A) and in most of these antheridial filaments after treatment with the JQ1 synchronous differentiation was observed, because spermatids mostly at three consecutive spermiogenesis stages were visible. However, some antheridia antheridial filaments at different distant developmental stages were revealed. The pattern of such asynchronous differentiation was observed both after 24 h (stages II, V, X; Figure 2D,G,H) and 48 h (stages V, VII–X; Figure 2I) treatments with the inhibitor. Some spermatids at stage X looked differently after JQ1 (Figure 2F) than those in the control (Figure 2C). They were thin with an uneven surface.

### 3.2. Influence of JQ1 on Spermatids Ultrastructure During C. vulgaris Spermiogenesis

Ultrastructural changes in both nucleus and cytoplasm area were observed in most spermatids treated with JQ1 (24 h and 48 h) at a given stage in comparison to the control. In some spermatids, these disorders did not affect the whole cell, that is they were not observed in the nucleus but only in the cytoplasm.

TEM image analysis revealed in the treated material only the sporadic occurrence of the fusion of two spermatid nuclei at stage II (Figure 3C) with visible continuity of the nuclear envelope (Figure 3D). These single images were visible in 10 analysed antheridia. In the other treated spermatids at this stage (Figure 3B), the nuclei pattern was the same as in the control (Figure 3A).

In nuclei of certain spermatids at stages II–V, in both time variants of JQ1 treatment, condensed chromatin adhered to the nuclear envelope; however, non-condensed chromatin showed a looser configuration and in its area bright or empty spaces were visible (Figure 4B,C; Figure 5B–D; Figure 6C,D), but these were not present in the control (Figure 4A; Figure 5A; Figure 6A). These smaller or larger spaces, which are often surrounded by a clearly visible membrane that resembles nuclear reticulum (NR), were also visible (Figure 5B,C; Figure 6C). After 48 h treatment with JQ1 in the central part of spermatid nucleus, non-condensed chromatin in the form of flocculent clusters was seen, between which bright spaces deprived of nuclear material more numerous than after 24 h treatment were visible (Figure 4C; Figure 5D,G; Figure 6D).

In the spermatids after treatment with JQ1 (24 h) already from stage II, both broader endoplasmic reticulum (ER) cisternae and thicker nuclear envelope (Figure 4B; Figure 5B,C,F; Figure 6B,C) in comparison with the control material (Figure 4A; Figure 5A; Figure 6A) were visible. These extended and fused the ER cisternae system and nuclear envelopes were filled with darker contents than in the control. The NR structure, similar to the extended ER cisternae system, was present in the control material on the spermatid nucleus area not earlier than at stage V (Figure 6A). In addition to the ER area, ultrastructural changes could also be seen in another part of the cytoplasm, which was characterized by a significant reduction, which is particularly noticeable after prolonged JQ1 treatment (Figure 4C; Figure 5D,G; Figure 6D). In its area, there are a lot of ER vesicles and probably autolytic vacuoles containing fragments of cytoplasm, intended for digestion and also different autophagic bodies (Figure 4B,C; Figure 5C,D,F,G; Figure 6B–D).

At stage IV, spermatid nuclei shifted to one of the side cell walls and the condensed chromatin adhered to the nuclear envelope. The same image was characteristic of a very early transition stage between IV and V, at which ER cisternae appeared (Figure 5E). Among the antheridial filaments of stage IV treated with JQ1 (24 h), there were those in which the arrangement of the spermatid nuclei was typical of this stage, however, the chromatin system in these nuclei resembled a network-like structure (Figure 5F) that is characteristic of the control spermatids only at stage VI (Figure 7A). Accelerated reorganization of chromatin into a network-like structure was also observed in some stage III spermatids. In some spermatids at stages V (Figure 6C) and VI, NR was found in the nucleus in a much larger amount and it occupied a larger area compared to the control.

Analysis of micrographs revealed in the same antheridial filament sporadic side-by-side occurrence of spermatids at two different stages of spermiogenesis (VI and VIII) (Figure 7C).

At stage VI (JQ1 24 h), the whole area of spermatid nucleus was uniform and filled with short chromatin fibrils arranged in different directions, between which there were lighter areas (Figure 7C). It was difficult to distinguish here between condensed and non-condensed chromatin, which in the control created a characteristic network-like arrangement (Figure 7A). Chromatin was not disturbed in all spermatids at stage VI. Some spermatids in which disorders were only visible in the cytoplasm were observed (Figure 7B).

At stage VIII and the last spermiogenesis stages (IX, X), instead of parallelly arranged long chromatin fibrils (Figure 8A), condensing chromatin was observed in the form of shorter or longer bands or spots, between which there were areas partially lacking fibrils (Figure 8B,C,E–G).

These changes became more pronounced after 48-h treatment with JQ1 (Figure 9) and are visible both on longitudinal and cross sections (Figure 9B).

Similarly as before, the disturbed cytoplasm was visible. A lot of bright spaces and numerous autolytic vacuoles were found (Figure 8C–H; Figure 9) that were not present in the control material (Figure 8A). However, in the treated material there were also spermatids that had the correct chromatin structure, but a disturbed cytoplasm (Figure 8H). The changes presented above were also visible at the final stage of spermiogenesis (X). Spermatozoid in the control material had strongly condensed chromatin and reduced cytoplasm (Figure 10A,C), in contrast to that after treatment with JQ1 (Figure 10B,D).

At all stages after 48 h treatment with JQ1, lack of cohesion of cellular structures of spermatids could be seen; it was difficult to notice cell walls in the antheridial filament cells and sometimes to recognize the typical structure of these spermatids (Figure 4C; Figure 5D,G; Figure 6D; Figure 9C). However, inhibitor JQ1 did not change the spermatid nucleus shape and microtubular manchette.

## 4. Discussion

Spermatogenesis in the green algae *Chara* is the object of research that has been conducted for many years also in our department [1,19]. Previous studies showed that in spermiogenesis of this genus there were some similarities to this process in vertebrates, mainly mammals and invertebrates [3,19,28,30]. However, there is still little data in the literature on the role that bromodomains play in the process of spermatogenesis in algae, thus, in this paper, this issue is mainly discussed on the basis of animal material.

BRD proteins take part in many cellular processes. Studies concerning the contribution of bromodomains and their inhibitors to chromatin epigenetic modifications include not only sperm cells but also different cancer cell lines [22,37,38,39]. The latest research conducted on human hepatocyte carcinoma (Hep G2 cell line) concerned the influence of bromodomain inhibitor JQ1 on lipid homeostasis [40].

During spermiogenesis, spermatids undergo the transformation from somatic cells into generative ones. In *C. vulgaris*, antheridial filaments containing spermatids, which are developing inside a complicated cell complex, antheridium, are used as the research model; however, in animal studies, mainly cell lines are used. The correct differentiation of spermatids in *C. vulgaris* is a complex process that requires the interaction of many elements and participation of the chromatin remodeling factor. In this alga Brg1 protein, a functional homologue of Swi2/Snf2 catalytic subunit of the SWI/SNF chromatin remodeling complex was shown [19]. Western blot analysis revealed two isoforms of Brg1 in the antheridial filament cells extract, and immunofluorescent and immunogold studies showed the presence of this protein at spermiogenesis stages I–VII. At these stages, Brg1 coexists with histone H4 acetylation at the Lys 12 position [30], and similar to most plant proteins, contains one bromodomain at the C-terminus [7,41].

Bromodomains participate in the spermatogenesis control system [42]. Studies of this process using bromodomain inhibitors i.e., highly specific JQ1, showed that it suppressed the expression of many genes that play essential roles in germ cells [24,43,44]. This inhibitor does not affect the proliferative properties of spermatogonia, but influences later stages of their development, spermatocytes and spermatids [37]. JQ1 blocks the differentiation of mouse sperm cells [25,43] and causes a decrease in the number of round spermatids and spermatozoa [37].

In human cutaneous T-cell lymphoma cell lines, this inhibitor inhibits proliferation, retaining cells in the G0/G1 phase and reducing their number in the S phase. The impact of the inhibitor is stronger in interaction with the histone deacetylase inhibitor SAHA [39]. A similar cell cycle arrest was observed in mice glioma stem cells [45]. JQ1 inhibits tumor growth and induces apoptosis, which is used in the epigenetic therapy of cancer [39,44,46,47]. Some BET inhibitors have also been tested in the treatment of different rheumatic diseases [48].

In the case of the examined algae, the effect of JQ1 on the proliferative phase of spermatogenesis was not analysed. However, a significant percentage decrease in spermatid count at stage 64/sp and at the early stages of spermiogenesis I–III, which is similar to mouse round spermatids, was observed. In *Chara*, the cell cycle lacks G1 phase (type S + G2 + M). Based on the above information, it can be assumed that, as in cell lines, also in the case of this alga, JQ1 may inhibit proliferation and the reduction of S phase leads to decreased spermatid number at stages 64/sp–III. In addition, after treatment with JQ1 in the same antheridium, asynchronous spermatid differentiation was observed, which might be the result of changes in the earlier proliferative phase. Because previous studies revealed at stages 64/sp–III the most intensive positive immunoreaction with the anti-Brg1 antibody [19], it seems that the decrease in the spermatid number may result from disturbance in their differentiation caused by blocking the bromodomain function. There are no data concerning the JQ1 inhibitory effect on Brg1 protein in spermatids. Studies on pancreatic cancer cells showed that JQ1 inhibited Brg1-mediated functions [49].

The studies conducted on male reproductive cells of mammals (man and mouse) showed that bromodomain inhibitor JQ1 blocked BRDT, and the effects of its activity depended on the time of application and its dose. JQ1 affected the developmental stages of these cells already during meiosis and successive post-meiotic stages [43,50]. In the studied alga, it was not possible to examine how JQ1 treatment affected meiosis, because *Chara* has a different type of meiosis (to plants and mammals), which occurs only after fertilization. During spermiogenesis, after treatment with JQ1, a decrease in the number of sperm cells was accompanied by changes of their motility. As a result of a long-term daily treatment of mouse with JQ1, a reduced volume of testis and of seminiferous tubules area was observed. Disturbances during the developmental stages of male reproduction cells in mice did not concern all cells. These changes were visible on histochemical sections of testis tubules and finally led to the formation of incorrect round spermatids, as well as multinucleated symplasts [43].

In *C. vulgaris* spermatids after JQ1 treatment, ultrastructural changes concerning the whole cell area were not present in all cells. It is interesting that the fusion of two spermatid nuclei at stage II was also observed. Studies on root meristem cells of *Narcissus* revealed that this fusion might occur as a result of the incomplete primary septum formation, separating these two cells, in telophase [51]. It can be hypothesized that, just like multinucleated spermatids [52], this image in alga may indicate some degree of degeneration of spermatids caused by JQ1.

Studies on pancreas tumor cells showed that JQ1 caused DNA damage, the marker of which is γH2AX. The number of fluorescent foci of γH2AX increased in JQ1 in a dose-dependent manner [44,47]. At *C. vulgaris*, DNA double-stranded breaks occur at the middle stages of spermatid differentiation and are necessary for the correct rearrangement the chromatin structure from nucleosomal to that connected with protamines. Their lack leads to disturbances in nucleoprotein exchange and chromatin condensation at stages VI–VIII of spermiogenesis [3].

No ultrastructural studies have been conducted so far on spermatids treated with bromodomain inhibitors. On the TEM level, only the impact of a potent bromodomain inhibitor, I-BET-151, on changes in cardiomyocyte mitochondria of male rodents, was shown [22]. This paper is the first presentation of the ultrastructural analysis of the spermatids treated with JQ1 that showed disturbances in the chromatin system, already at the early spermiogenesis stages and lasting until its end. Simultaneously, a significant decrease in cytophotometrically measured Feulgen intensity at stage X was observed, which might be related to the degeneration of DNA content in these spermatids, thus leading to the formation of defective mature spermatozoids. An abnormal DNA content, which is associated with disturbed chromatin condensation, was also revealed in some spermatids in mice treated with different insecticides. Studies on spermatozoa of infertile animals and humans showed that DNA values were either much lower or higher in comparison to fertile ones [53].

The prolonged treatment with JQ1 (48 h) led to the formation of spermatids altered to such an extent that it was sometimes difficult to recognize individual cell structures. Compared to the earlier research [3,28], such pronounced changes in the structure of chromatin and the cytoplasm of spermatids have not been observed so far. JQ1 blocks binding bromodomain to acetylated histones, and this is why changes in the chromatin condensation could be caused by a disturbance in the exchange of nuclear proteins. An interesting TEM observation concerns the ER system in spermatids. During *C. vulgaris* spermiogenesis, ER is most developed at the advanced stage V [1]; however, after JQ1 treatment already at the earlier stages of spermiogenesis (II–IV), both distended ER cisternae and the nuclear envelope connected to them were visible. Since ER plays an essential role in cellular processes, primarily in protein assembly, it can be assumed that the observed images were caused by a disturbance in the proper functioning of this cellular domain as a result of JQ1. Studies conducted on different human cancer cell lines showed that BET inhibitors i.e., JQ1, induced apoptosis [39,44,54]. It was also revealed that this bromodomain inhibitor might induce ER stress, which then leads to apoptosis [55]. Electron microscopic observations showed thicker and fused ER in fibroblasts [56], dilated ER in rat in diaphragm muscle [57], and hepatocytes [58] also as a result of ER stress, similarly as in *Chara*. In internodal cells of *C. vulgaris* thallus, the action of heavy metals (mercury) also caused the appearance of dilated ER cisternae [59]. In root tips of *Arabidopsis* vacuoles indicating ER stress-dependent autophagy [60] and in human spermatozoa autophagy vesicles under stressful conditions [61] were observed. On the basis of the above information and the presented TEM image analysis, it can be assumed that the disturbances in ER system and formation of numerous autolytic vacuoles in the cytoplasm of spermatids was the result of JQ treatment. It is known that ER is a site of protamine synthesis [36], and as a result of activation of ER stress response factors, further translation is inhibited [62,63]. Probably the currently observed changes in algae are also associated with the inhibition of the ubiquitin/26S proteasome system, which plays an important role in the spermiogenesis of *C. vulgaris*. Thus, its dysfunction makes the proper exchange of histones into protamines impossible [28]. Studies on mouse spermatids revealed that a lack of the proteasome activator PA200 only delayed nucleohistone disappearance [64]; however, under the influence of JQ1, it can be expected that the function of this activator, containing a bromodomain-like region, will be blocked.

The current studies broaden the knowledge about algae Charophyta spermiogenesis. The results obtained revealed the essential role of bromodomains in this process. The blockage of their proper functioning caused such pronounced disturbances that have not been seen so far after treatment with different inhibitors. These changes both in the nucleus and cytoplasm of spermatid may confirm that JQ1 inhibitor interferes with the spermatid differentiation on many interdependent levels. This issue has not been analyzed in detail on the ultrastructural level until now.

## Figures and Tables

**Figure 1 cells-09-01352-f001:**
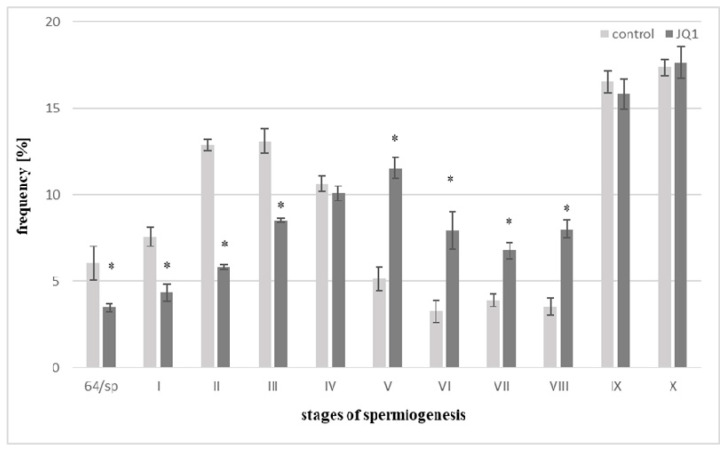
The percentage of *C. vulgaris* spermatids during the transitory stage between proliferation phase and spermiogenesis (64/sp) and spermiogenesis stages (I–X) in the control and after 24-h treatment with JQ1, a bromodomain inhibitor. Error bars represent ± SD. Statistical results were analyzed by Student *t*-test with the use of Microsoft Excel 2000. Differences in the percentage of spermatids during consecutive stages between control and JQ1-treated plants are statistically significant, *p* ≤ 0.05. In this experiment, three replicates were performed per each variant (the control and bromodomain inhibitor). * These differences are statistically significant in comparison with the control.

**Figure 2 cells-09-01352-f002:**
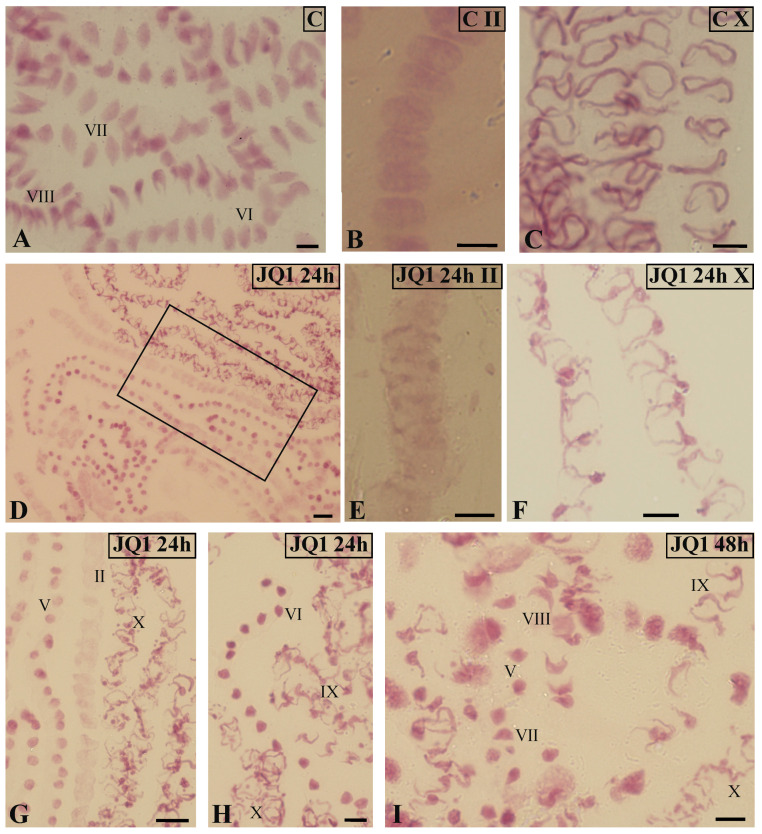
Feulgen-stained spermatid nuclei at antheridial filaments of *C. vulgaris* spermiogenesis. (**A**–**C**) the control material at early (II; **B**) and late (X; **C**) stages; (**A**) synchronous differentiation of spermatids, VI–VIII consecutive spermiogenesis stages; (**D**–**I**) material after 24-h (**D**–**H**) and 48-h (**I**) treatment with JQ1; antheridial filaments at different distant developmental stages (II, V, IX, X); (**G**) magnified area from (**D**); (**E**,**F**) spermatids at early (II; **E**) and late (X; **F**) stages; individual images in the frame treatment, time of treatment, and stage of spermiogenesis were given; Scale bars = 5 μm (**A**–**C**,**E**–**I**), 10 μm (**D**).

**Figure 3 cells-09-01352-f003:**
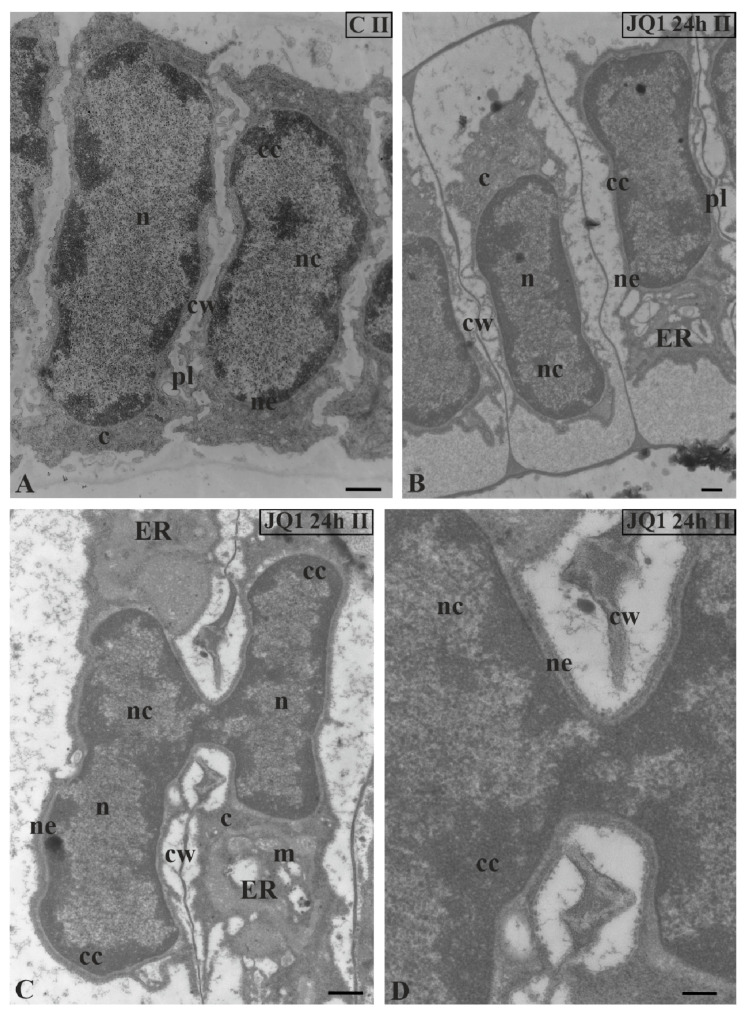
The effect of JQ1 on *C. vulgaris* spermatids at stage II after 24 h treatment. (**A**) the control; (**B**–**D**) after treatment with JQ1; (**C**) The fusion of two spermatid nuclei on a longitudinal section; (**D**) magnification of the fusion site with visible continuity of the nuclear envelope and broken cell wall between spermatids; c, cytoplasm; cc, condensed chromatin; cw, cell wall; ER, endoplasmic reticulum; m, mitochondrion; n, nucleus; nc, non-condensed chromatin; ne, nuclear envelope; on individual images in the frame treatment, time of treatment and stage of spermiogenesis were given; Scale bars = 500 nm (**A**–**C**), 200 nm (**D**).

**Figure 4 cells-09-01352-f004:**
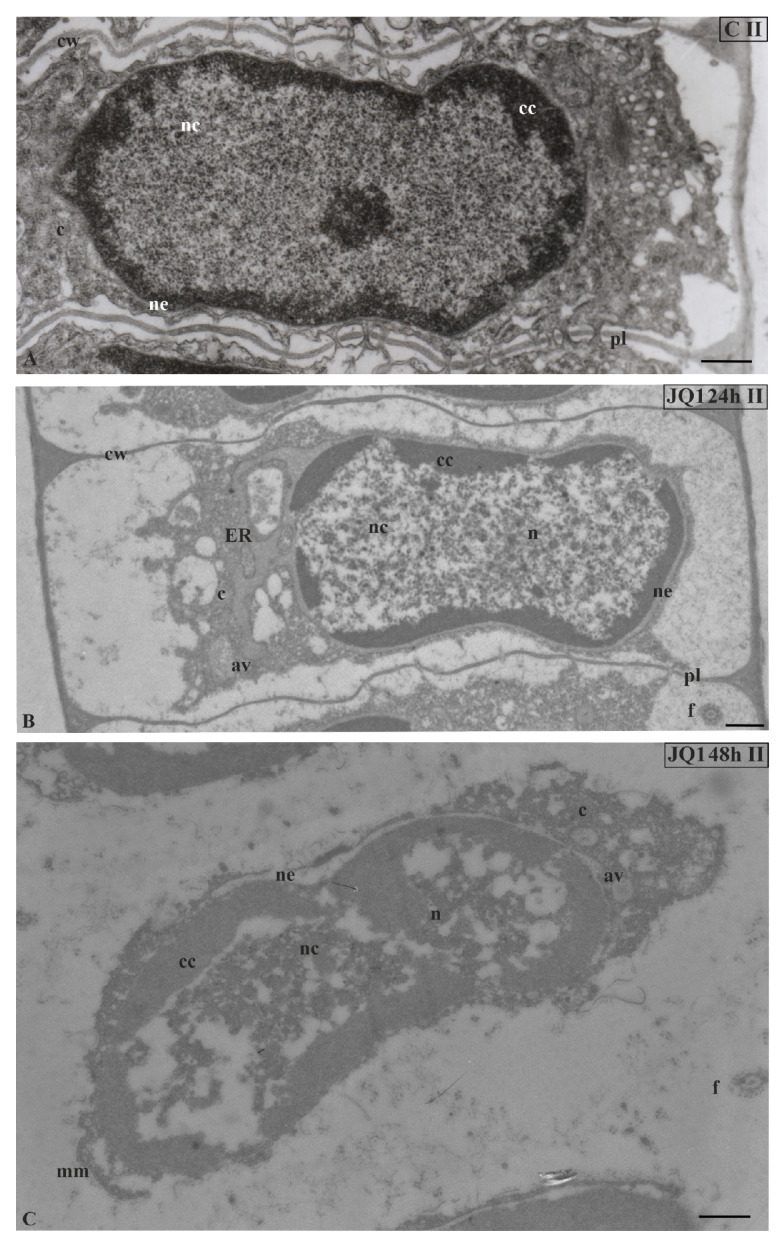
Ultrastructural changes in *C. vulgaris* spermatids at stage II; (**A**) the control; (**B**) after 24-h and (**C**) 48-h treatment with JQ1; changes in non-condensed chromatin and cytoplasm structure. In spermatids dilated nuclear envelope together with ER cisternae (**B**) and in nucleus loose spaces (**B**,**C**) are visible. Longitudinal section of spermatids; av, autolytic vacuole; c, cytoplasm; cc, condensed chromatin; cw, cell wall; ER, endoplasmic reticulum; f, flagellum; mm, microtubular manchette; n, nucleus; nc, non-condensed chromatin; ne, nuclear envelope; pl, plasmodesmata; on individual images in the frame treatment, time of treatment and stage of spermiogenesis were given; Scale bar = 500 nm.

**Figure 5 cells-09-01352-f005:**
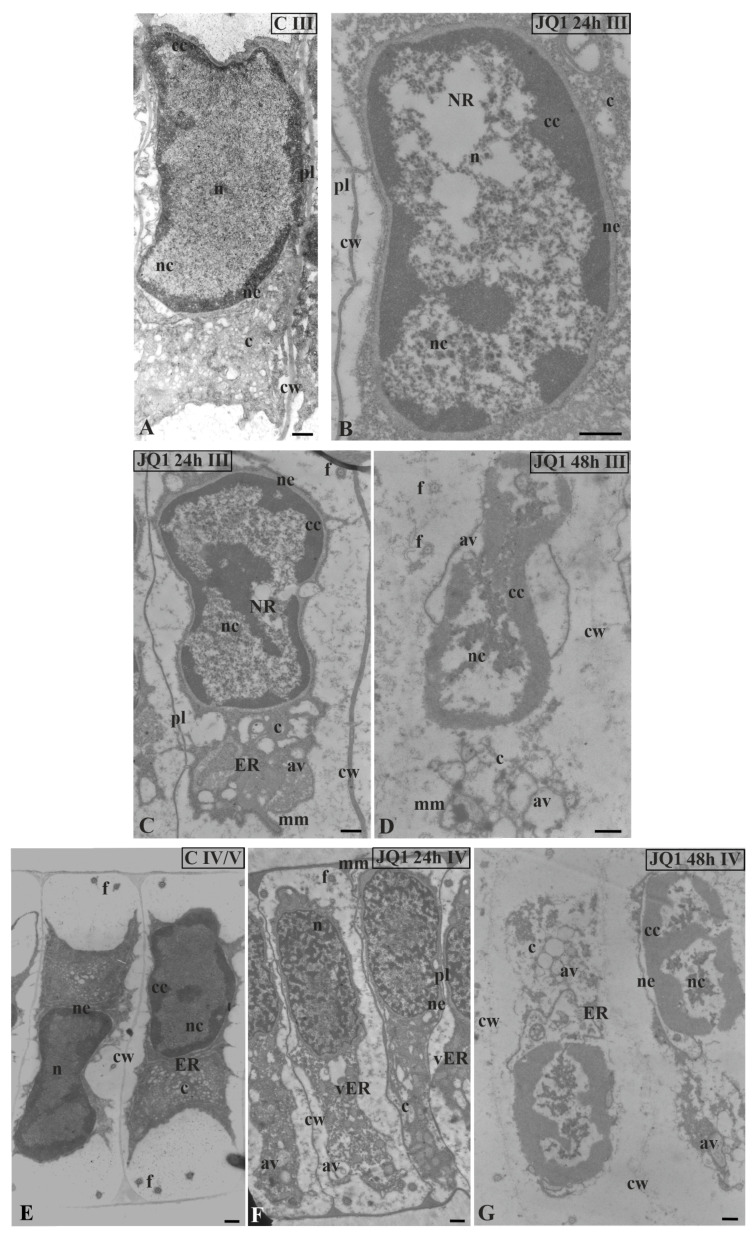
The effect of JQ1 on *C. vulgaris* spermatids ultrastructure at stage III (**A**–**D**) and IV (**E**–**G**); (**A**,**E**) the control; (**B**–**F**) after 24 h (**B**,**C**,**F**) and 48-h (**D**,**G**) treatment with JQ1; extended nuclear envelop (**B**,**C**) and bright spaces in nucleus in non-condensed chromatin (**B**–**D**,**G**), these spaces cover almost the whole central part of the nucleus (**D**); (**C**) reduced cytoplasm with dilated ER cisternae; (**D**,**G**) barely visible cell wall between spermatids, strongly reduced cytoplasm, ER cisternae not visible, and a lot of vesicles and lytic vacuoles are present; (**E**–**G**) nuclei shifted to one of the side walls, net-like chromatin pattern (**F**) is not typical of this stage. Longitudinal section of spermatids; av, autolytic vacuole; c, cytoplasm; cc, condensed chromatin; cw, cell wall; ER, endoplasmic reticulum; f, flagellum; mm, microtubular manchette; n, nucleus; nc, non-condensed chromatin; ne, nuclear envelope; NR, nuclear reticulum; pl, plasmodesmata; vER, endoplasmic reticulum vesicle; on individual images in the frame treatment, time of treatment, and stage of spermiogenesis were given; Scale bar = 500 nm.

**Figure 6 cells-09-01352-f006:**
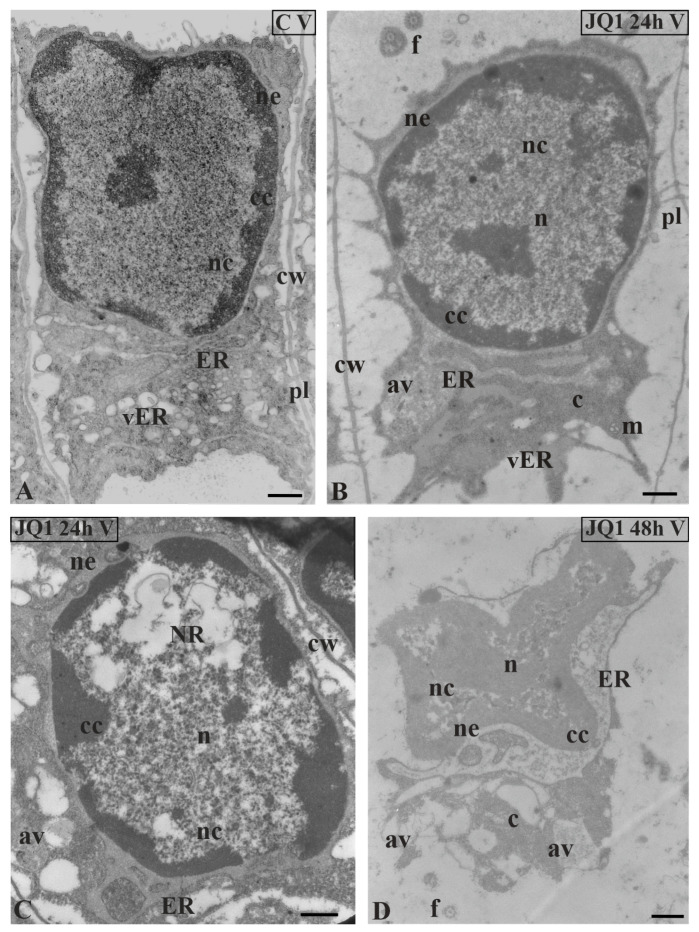
Ultrastructural changes in *C. vulgaris* spermatids at stage V; (**A**) the control; (**B**,**C**) after 24 h and (**D**) 48 h treatment with JQ1; (**B**) Minor changes in non-condensed chromatin, however, strongly dilated ER cisternae and nuclear envelope are visible; (**C**,**D**) progressive changes in the nucleus, and the presence of NR and numerous bright spaces; in cytoplasm, the presence of swollen ER cisternae, many vesicles and autolytic vacuoles. Longitudinal section of spermatids; av, autolytic vacuole; c, cytoplasm; cc, condensed chromatin; cw, cell wall; ER, endoplasmic reticulum; f, flagellum; m, mitochondrion; n, nucleus; nc, non-condensed chromatin; ne, nuclear envelope; NR, nuclear reticulum; pl, plasmodesmata; vER, endoplasmic reticulum vesicle; on individual images in the frame treatment, time of treatment and stage of spermiogenesis were given; Scale bar = 500 nm.

**Figure 7 cells-09-01352-f007:**
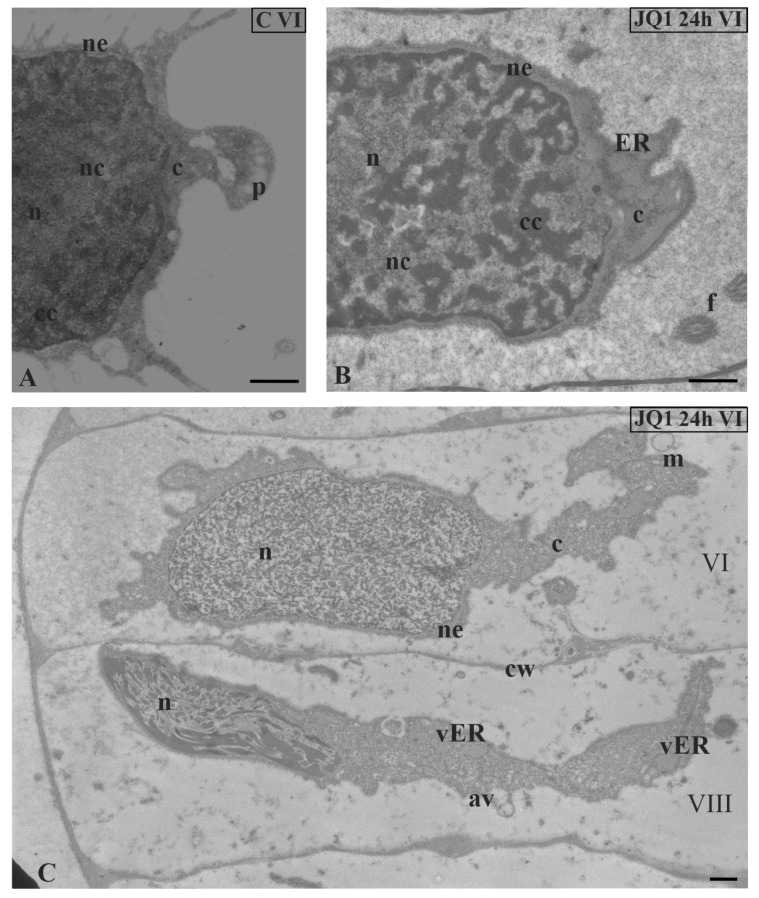
The effect of JQ1 on *C. vulgaris* spermatids ultrastructure at stage VI; (**A**) the control, nucleus with net-like chromatin structure; (**B**,**C**) after 24-h treatment with JQ1; (**B**) chromatin structure as in the control, swollen nuclear envelope and ER cisternae; (**C**) spermatids in the same antheridial filaments at distant developmental stages (VI and VIII); disturbed net-like chromatin structure in nucleus at stage VI and longer loosely arranged chromatin fibrils with spaces between them at stage VIII. Longitudinal section of spermatids; av, autolytic vacuole; c, cytoplasm; cc, condensed chromatin; cw, cell wall; ER, endoplasmic reticulum; f, flagellum; m, mitochondrion; n, nucleus; nc, non-condensed chromatin; ne, nuclear envelope; p, plastid; vER, endoplasmic reticulum vesicle; on individual images in the frame treatment, time of treatment and stage of spermiogenesis were given; Scale bar = 500 nm.

**Figure 8 cells-09-01352-f008:**
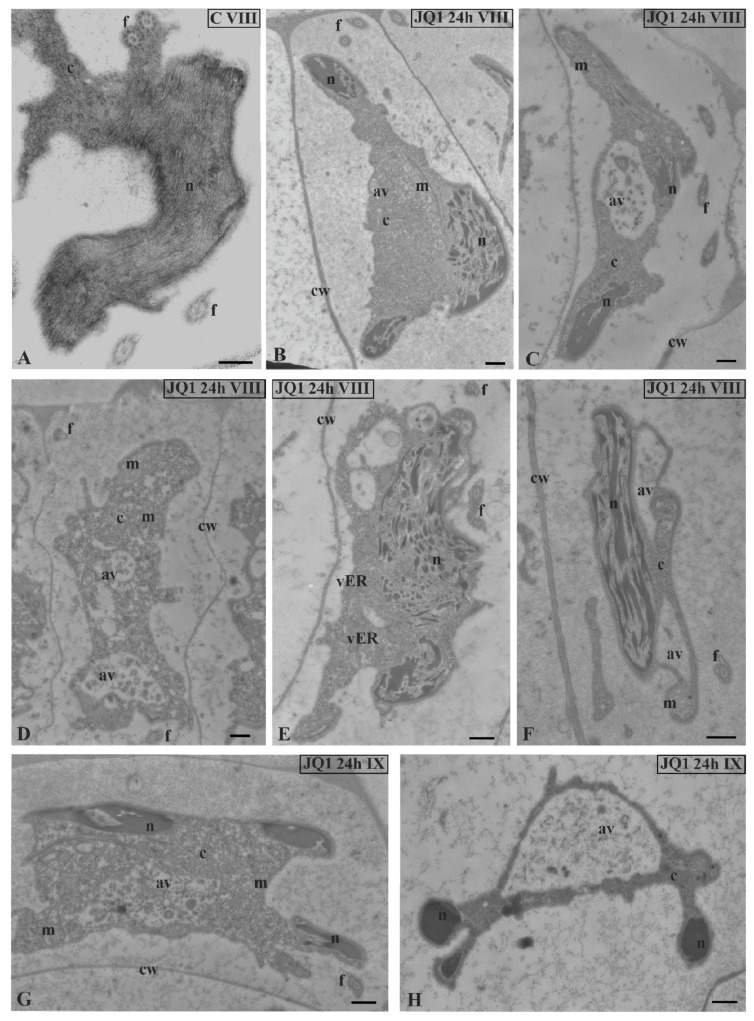
The effect of JQ1 on *C. vulgaris* spermatid ultrastructure at stages VIII (**B**–**F**) and IX (**G**,**H**) after 24-h JQ1 treatment; (**A**) the control, nucleus with long parallel chromatin fibrils; (**B**,**C**,**E**–**G**) disturbances in chromatin structure, and shorter or longer chromatin fibrils in the form of clusters separated by brighter areas; (**C**–**H**) changes in the cytoplasm area, numerous vesicles and large autolytic vacuoles containing fragments of cytoplasm; (**H**) spermatid with correctly condensed chromatin but with changes in cytoplasm. Longitudinal section of spermatids. Nucleus on cross-section (**H**); av, autolytic vacuole; c, cytoplasm; cw, cell wall; f, flagellum; m, mitochondrion; n, nucleus; vER, endoplasmic reticulum vesicle; on individual images in the frame treatment, time of treatment and stage of spermiogenesis were given; Scale bar = 500 nm.

**Figure 9 cells-09-01352-f009:**
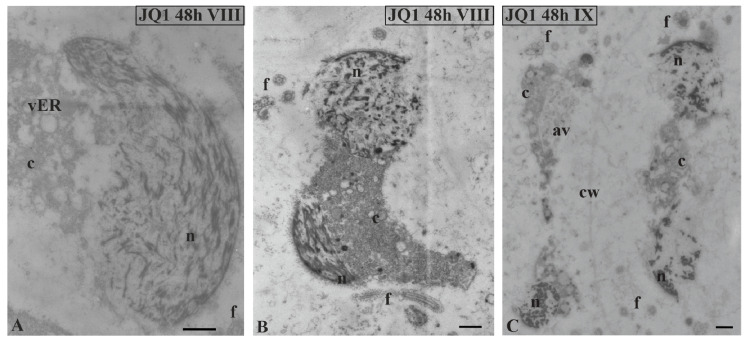
Ultrastructural changes in *C. vulgaris* spermatids after 48-h JQ1 treatment at stages VIII (**A**,**B**) and IX (**C**); (**A**–**C**) short chromatin fibrils with loose arrangement and bright spaces, these disturbances are visible on longitudinal and cross sections of the nucleus; cytoplasm with numerous vesicles and autolytic vacuoles containing fragments of cytoplasm, there is a barely visible cell wall between spermatids. Longitudinal section of spermatids; av, autolytic vacuole; c, cytoplasm; cw, cell wall; f, flagellum; n, nucleus; vER, endoplasmic reticulum vesicle; on individual images in the frame, time of treatment and stage of spermiogenesis were given; Scale bar = 500 nm.

**Figure 10 cells-09-01352-f010:**
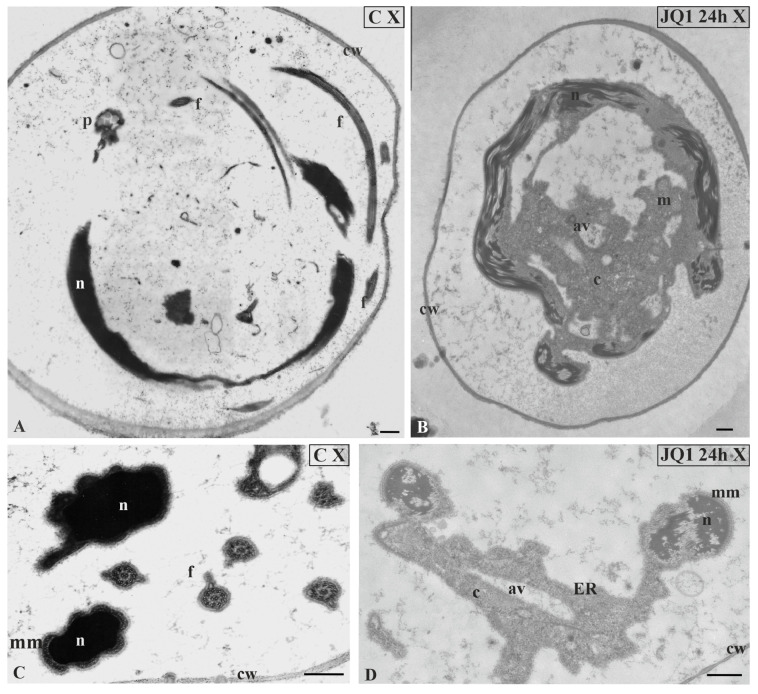
The effect of JQ1 (24 h) on *C. vulgaris* spermatozoids ultrastructure at stages X; (**A**,**C**) the control, very dense chromatin and reduced cytoplasm closely adjacent to the nucleus; (**B**,**D**) after treatment with JQ1, disturbances in chromatin structure, long bands chromatin fibrils loosely arranged with light spaces, cytoplasm with vesicles and autolytic vacuoles. Nuclei on longitudinal (**A**,**B**) and cross (**C**,**D**) section; av, autolytic vacuole; c, cytoplasm; cw, cell wall; ER, endoplasmic reticulum; f, flagellum; m, mitochondrion; mm, microtubular manchette; n, nucleus; p,plastid; on individual images in the frame were placed treatment, time of treatment on individual images in the frame treatment, time of treatment, and stage of spermiogenesis were given; Scale bar = 500 nm.

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
