# Peer review of "Blocking the Bromodomains Function Contributes to Disturbances in Alga Chara vulgaris Spermatids Differentiation"

_cells, 2020, doi:10.3390/cells9061352_

Round 1
Reviewer 1 Report
In this study, Agnieszka Wojtczak investigated the effect of JQ1, a bromodomain inhibitor, on the Alga Chara vulgaris Spermatids differentiation through Feulgen staining and analyzed in electron microscope. The author concluded that JQ1 incubation led to changes in spermatid number, disturbances in chromatin condensation and cytoplasm reduction. Though the study did not show evidences at molecular level, the detailed examination on subcellular and ultrastructural level did provide hint for the potential function of plant bromodomain in chromatin remodeling and Alga Spermatids differentiation, which match the scope of Cells journal. However, the manuscript need to be further improved addressing the following suggestions and questions.
In Introduction, lines 62-63, “Its biochemically active stereoisomer, (+)-JQ1 has potent, more specific action to BRD4 protein than among others to BRD2”, need to be rewritten. Does the author means that JQ1 is more potential to inhibit BRD4 as compare to other ones (BRD2 and BRD4 et al) in animals?
In Introduction, lines 63-65, the loss of first bromodomain in which gene?
In Introduction, Does any study report how many or which proteins containing bromodomain were identified in Alga? If so, please give a brief introduction.
In methods, lines 92-94, please give a brief explanation, for JQ1 treatment, why the author select the concentration 100 um and incubation time 24 and 48 hour? Any reference for this treatment?
In methods, line 96, line 99 and line 104, please clarify it is JQ1-treated material instead of the incubated material.
In Results, Figure 1, please clarify how many biological replicates were performed for the t-test, and how to explain the frequency of spermatids in stages V-VIII less than I-III in Control?
In Results lines 62-65, the first sentence is confusing, Why the author said “Ultrastructural changes were not observed in all spermatids at a given stage”? Does the author means “Ultrastructural changes after JQ1 treatment were observed in some of (but not all of) the spermatids at a given stage”? If so, what are the percentage of cells affected by the JQ1 in each stage? And what does “Some spermatids, however, maintained the correct image” means?
In Figure 3, why the image of DMSO-treated material not shown as control? How it would looks like in normal condition?
In Discussion line 307-308, the author point out that there is a Brg1 protein in this alga. Does any data show the expression of Brg1 gene in each stage of spermiogenesis?
Reviewer 2 Report
The author detected the consequences of bromodomain inhibitor JQ1 treatment on algae Chatophyta spermiogenesis. These changes were analyzed on the ultrastructural level and led to spermatid degeneration. This topic is interesting and Ms. has certain scientific value. However, I have a few concerns and questions about the study and they are as follows:
There is no mention of how the concentration of inhibitor was chosen, It is not clear, whether studied structural changes are a consequence of bromodomain inhibition or potential cytotoxic effect of used concentration, or how other cell types can survive after this treatment.
The main conclusions should be supported by statistic, e.g. fusion of spermatid nuclei after JQ1 treatment (this statement is not clear based on line 166, the percentage of fused nuclei should be presented). All section with statistical methods should be added or improved in Figure legend; the use of replicates, the number of spermatids in each stage of spermiogenesis quantified by comparative analysis needs to be presented (Figure 1).
The Results are not described in a clear way. The Figures should be reconsidered and significantly improve their labeling, especially in the case of Figures consisting of more than 3 scans. (e.g. add the description of individual scan from microscope above: treatment, time of treatment, and stage of spermiogenesis or improve the figure legend. Thus it is difficult for readers to understand the text).
Minor points:
Line 37: acetyltransferases instead acetylotransferase
Why is used both (+)-JQ1 vs. JQ1?
Line 134: Error bars instead of bars.
Line 143: For easier orientation, I would prefer adding Figure 2C vs. 2F.
Line 178: Nuclear reticulum is not labeled in Figure 5B, as mentioned here.
In Figure 4A – Labeling “nc” is invisible.
In Figure 4C – The explanation of labeling “mm” is missing.
Figure 5: Control of stage IV is missing. Next, the cell wall is not labeled in Figure 5D.
Line 247: The percentage of spermatids with disturbed and not disturbed chromatin structure would be useful.
Author Response
.

Round 2
Reviewer 1 Report
The authors address all of my comments.
Author Response
Replies to Reviewer 1 comments:
We thank this reviewer.
A language correction was carried out.
The changes in the manuscript were highlighted with the use "Track Changes" function in Microsoft Word. Number of lines is according to earlier submitted manuscript
Reviewer 2 Report
The majority of concerns raised in the 1st round of review have been properly addressed by the author. However, the proof of the JQ1 inhibitory effect on inhibition BRD protein (Brg1) in this model system is missing.
In Introduction, line 37, why do you keep in the text histone acetyltransferase instead histone acetyltransferases? Please, use the plural.
In Results, lines 216-219, please use correct links to figures 5E-F. Also, explain why instead control IV you show control IV/V.
Author Response
We thank this reviewer, please see the attachment.
